

# Integrative analysis of Iso-Seq and RNA-seq data reveals transcriptome complexity and differentially expressed transcripts in sheep tail fat

Zehu Yuan[1], Ling Ge[2], Jingyi Sun[3], Weibo Zhang[2], Shanhe Wang[2], Xiukai Cao[1] and Wei Sun[1,2]

[1] Joint International Research Laboratory of Agriculture and Agri-Product Safety of Ministry of Education, Yangzhou University, Yangzhou, China
[2] College of Animal Science and Technology, Yangzhou University, Yangzhou, China
[3] College of Veterinary Medicine, Yangzhou University, Yangzhou, China

## ABSTRACT

**Background:** Nowadays, both customers and producers prefer thin-tailed fat sheep. To effectively breed for this phenotype, it is important to identify candidate genes and uncover the genetic mechanism related to tail fat deposition in sheep. Accumulating evidence suggesting that post-transcriptional modification events of precursor-messenger RNA (pre-mRNA), including alternative splicing (AS) and alternative polyadenylation (APA), may regulate tail fat deposition in sheep. Differentially expressed transcripts (DETs) analysis is a way to identify candidate genes related to tail fat deposition. However, due to the technological limitation, post-transcriptional modification events in the tail fat of sheep and DETs between thin-tailed and fat-tailed sheep remains unclear.

**Methods:** In the present study, we applied pooled PacBio isoform sequencing (Iso-Seq) to generate transcriptomic data of tail fat tissue from six sheep (three thin-tailed sheep and three fat-tailed sheep). By comparing with reference genome, potential gene loci and novel transcripts were identified. Post-transcriptional modification events, including AS and APA, and lncRNA in sheep tail fat were uncovered using pooled Iso-Seq data. Combining Iso-Seq data with six RNA-sequencing (RNA-Seq) data, DETs between thin- and fat-tailed sheep were identified. Protein protein interaction (PPI) network, Gene Ontology (GO) and Kyoto Encyclopedia of Genes and Genomes (KEGG) enrichment analyses were implemented to investigate the potential functions of DETs.

**Results:** In the present study, we revealed the transcriptomic complexity of the tail fat of sheep, result in 9,001 potential novel gene loci, 17,834 AS events, 5,791 APA events, and 3,764 lncRNAs. Combining Iso-Seq data with RNA-Seq data, we identified hundreds of DETs between thin- and fat-tailed sheep. Among them, 21 differentially expressed lncRNAs, such as ENSOART00020036299, ENSOART00020033641, ENSOART00020024562, ENSOART00020003848 and 9.53.1 may regulate tail fat deposition. Many novel transcripts were identified as DETs, including 15.527.13 (*DGAT2*), 13.624.23 (*ACSS2*), 11.689.28 (*ACLY*), 11.689.18 (*ACLY*), 11.689.14 (*ACLY*), 11.660.12 (*ACLY*), 22.289.6 (*SCD*), 22.289.3 (*SCD*) and 22.289.14 (*SCD*). Most of the identified DETs have been enriched in GO

Corresponding author
Wei Sun, dkxmsunwei@163.com

and KEGG pathways related to extracellular matrix (ECM). Our result revealed the transcriptome complexity and identified many candidate transcripts in tail fat, which could enhance the understanding of molecular mechanisms behind tail fat deposition.

# INTRODUCTION

Post-transcriptional modification events of precursor-messenger RNA (pre-mRNA), including alternative splicing (AS) and alternative polyadenylation (APA), have attracted an increasing interest (*Baralle & Giudice, 2017*; *Di Giammartino, Nishida & Manley, 2011*; *Gruber & Zavolan, 2019*; *Kornblihtt et al., 2013*; *Wang et al., 2015*). In livestock, accumulated evidence suggests that AS (*Fang et al., 2020*; *Leal-Gutierrez, Elzo & Mateescu, 2020*; *Xiang et al., 2018*; *Yuan et al., 2021*) and APA (*Deng et al., 2020*; *Jin et al., 2021*) could contribute to the formation of important economic traits. For example, both AS (*Motter, Corva & Soria, 2021*) and APA (*Nattrass et al., 2014*) in calpastatin (*CAST*) were associated with meat tenderness in beef. RNA-sequencing (RNA-seq) is a way to accurately quantify the AS events, *e.g.*, using the intron excision ratio (*Li et al., 2018c*). Unfortunately, it is unlikely to obtain the full-length transcripts and quantify the transcripts by RNA-seq because it is difficult to assemble full-length transcripts based on short reads and to exactly assign a short read to a certain transcript (*Kovaka et al., 2019*), which hamper the discovery of post-transcriptional events. The third-generation sequencing technology, such as PacBio isoform sequencing (Iso-Seq), which can directly produce full length transcripts (*Rhoads & Au, 2015*), providing an opportunity to discover novel genes and novel isoforms (*Au et al., 2013*; *Beiki et al., 2019*; *Chen et al., 2017*; *Karlsson & Linnarsson, 2017*) and to investigate the transcriptome complexity in mammals (*Deng et al., 2020*; *Li et al., 2018b*). In addition, the results from the previous study suggested that integrative analysis of Iso-Seq and RNA-seq data could accurately quantify the abundance of transcripts (*Chao et al., 2019*; *Li et al., 2017*). Thus, using Iso-Seq and RNA-seq could accurately identify the post-transcriptional modification events and differential expressed isoforms.

Hu sheep, a short fat-tailed sheep breed, is one of the most popular sheep breeds in China because of its high fecundity. In an intensive production system, Hu sheep is usually used as ewe crossing with elite ram breeds, *e.g.*, Dorper, to produce meat. Backcross ((Dorper × Hu) × Hu, DHH) and grading up (Dorper × (Dorper × Hu), DDH) are two common hybridization methods in sheep meat production. The weight of the tail fat of their hybrid offspring is positively related to the composition of the Hu sheep genome. Nowadays, both customers and producers prefer a small amount of tail fat. To breed for this phenotype, it is important to identify the candidate genes linked with tail fat and understand the underlying mechanisms of fat deposition.

**Table 1 Sample information.**

| Cross type | Sample size | Live weight (kg) | Carcass weight (kg) | Tail fat (kg) |
|---|---|---|---|---|
| Backcross ((Dorper × Hu) × Hu sheep, DHH) | 3 | 41.53 ± 0.95 | 23.70 ± 0.64 | 0.78 ± 0.21 |
| Grading up (Dorper × (Dorper × Hu sheep), DDH) | 3 | 40.80 ± 4.57 | 22.00 ± 2.27 | 0.11 ± 0.03 |
| T-test $t$ value | | 0.27 | 1.27 | 5.44 |
| T-test $df$ | | 2.17 | 2.32 | 2.10 |
| T-test $P$ value | | 0.8094 | 0.3161 | 0.0288 |

Many studies have attempted to identify the candidate gene associated with tail fat and understand the underlying mechanisms of fat deposition (*Ahbara et al., 2018*; *Amane et al., 2020*; *Baazaoui et al., 2021*; *Bakhtiarizadeh & Alamouti, 2020*; *Dong et al., 2020*; *Han et al., 2021*; *Li et al., 2018a*; *Mastrangelo et al., 2019a*; *Mastrangelo et al., 2019b*; *Moioli, Pilla & Ciani, 2015*; *Moradi et al., 2012*; *Wei et al., 2015*; *Xu et al., 2017*; *Yuan et al., 2017*; *Zhang et al., 2019*; *Zhao et al., 2020*; *Zhi et al., 2018*; *Zhu et al., 2016*). Previous studies suggested that microRNAs (*Miao et al., 2015a*; *Pan et al., 2018*), lncRNAs (*Bakhtiarizadeh & Salami, 2019*; *He et al., 2020*) and mRNAs (*Bakhtiarizadeh et al., 2019*; *Kang et al., 2017a*; *Miao et al., 2015b*; *Wang et al., 2014*) may regulate tail fat deposition in sheep. These results contribute to understanding the underlying mechanisms of fat deposition. However, due to the technological limitation, post-transcriptional modification events (*e.g.*, AS and APA) in the tail fat of sheep remains unclear, which might play an important role in tail fat deposition. In the current study, Iso-Seq was used to uncover post-transcriptional modifications in sheep tail fat and combined with RNA-Seq to investigate differential expressed transcripts (DETs) in tail fat between DHH and DDH. Our results could reveal transcriptome complexity in sheep tail fat and characterize DETs between DHH and DDH sheep.

## MATERIALS & METHODS

### Animal tissues collection and RNA extraction

After wearing (2-month old), all animals were fed to 6-month old in house with complete formula granulated feed (DafengGe Feed Technology Co., Ltd.) at Suyang Sheep Industry Co., Ltd (Fengxian, Jiangsu, China). All animals were eating and drinking freely. In total, six unrelated 6-month old male sheep (DDH = 3, DHH = 3) with similar live weight (Table 1) were selected to slaughter for sampling. Animal experiments were approved by the Experimental Animal Ethical Committee of Yangzhou University (NO.202103294).

After slaughter, tail fat tissues were sampled within 30 mins. Each fat sample was packed into a 1.5 ml cryotube. All the fat samples were quickly frozen in liquid nitrogen and stored at −80 °C. Total RNA was extracted from the tail fat using Trizol reagent Kit (TaKaRa, USA). The extraction protocol strictly follows the instructions of Kit. The quality of RNA samples was evaluated by Nanodrop 2,000 (Thermo Scientific™) and 2,100 Bioanalyzer (Agilent Technologies, Waldbronn, Germany). After quality control, all

the RNA samples with RNA Integrity Number (RIN) greater than 7.0 and 28S/18S ratio greater than 1.0 were used for sequencing.

## Library construction and RNA-sequencing

Library construction and sequencing of RNA-seq of six RNA samples were performed by a commercial sequencing service company (Frasergene Technology Co., Ltd, Wuhan, China). Briefly, the MGIEasy RNA Directional Library Prep Kit (MGI) and 1 μg of total RNA were used for library construction. Then, the library was sequenced on the MGISEQ-2000 platform. All sequencing data have been deposited in National Center for Biotechnology Information (NCBI) Short Read Archive (SRA) database and can be accessed under the BioProject accession number PRJNA745517.

## Iso-Seq library construction and sequencing

Six RNA samples (DDH = 3, DHH = 3) were pooled in equal quantities for Iso-Seq. Library preparation and Iso-Seq were performed by a commercial sequencing service company (Frasergene Technology Co., Ltd, Wuhan, China). Briefly, the SMARTer™ PCR cDNA synthesis kit (Takara Biotechnology, Dalian, China) was used to reverse-transcribed RNA into cDNA. Then, PCR Amplification was implemented by using 12–14 PCR cycles. PCR products were purified by the AMPure PB magnetic beads (Beckman Coulter, CA, USA). The BluePippin™ Size Selection System (Sage Science, MA, USA) was used for the size selection (>1 kb). Pacific Biosciences DNA Template Prep Kit 2.0 (Pacific Biosciences, CA, USA) was used for SMRTbell library construction. Agilent Bioanalyzer 2,100 system (Agilent Technologies, CA, USA) and Qubit fluorometer 2.0 (Life Technologies, CA, USA) were used for quality accession of the library. Then, the library was sequenced on the PacBio sequencing platform with 10 h sequencing movies.

## Iso-Seq data processing

The pipeline of bioinformatics analysis was shown in Fig. 1. The raw reads of Iso-Seq were preprocessed using SMRT Link v8.0 (https://www.pacb.com/wp-content/uploads/SMRT-Link-Release-Notes-v8.0.pdf). The adapter of polymerase reads was remove to get Subreads. Circular Consensus sequencing (CCS) reads were obtained from the Subreads using the following parameters: minimum subread length = 50, maximum subread length = 15,000, minimum number of passes = 3, and minimum predicted accuracy = 0.99. Then, CCS were classified into full length reads by lima software. Full-length non-chimeric reads (FLNCs) were full-length CCS reads with 5′ and 3′ cDNA primers and polyA. The LoRDEC v0.9 software (*Salmela & Rivals, 2014*) was used for FLNC correction using the high-quality short reads with the default parameters. The quality of RNA-seq data was examined using SOAPnuke v2.1.0 (*Chen et al., 2018*). FLNCs were aligned to the sheep reference genome (http://asia.ensembl.org/Ovis_aries_rambouillet/Info/Index) using GMAP (*Wu & Watanabe, 2005*).

## Novel genes and isoforms identification

Based on the results from alignments, FLNCs sharing the same splicing event were merged into one isoform. The isoforms with 5′ terminal region degraded were excluded for further

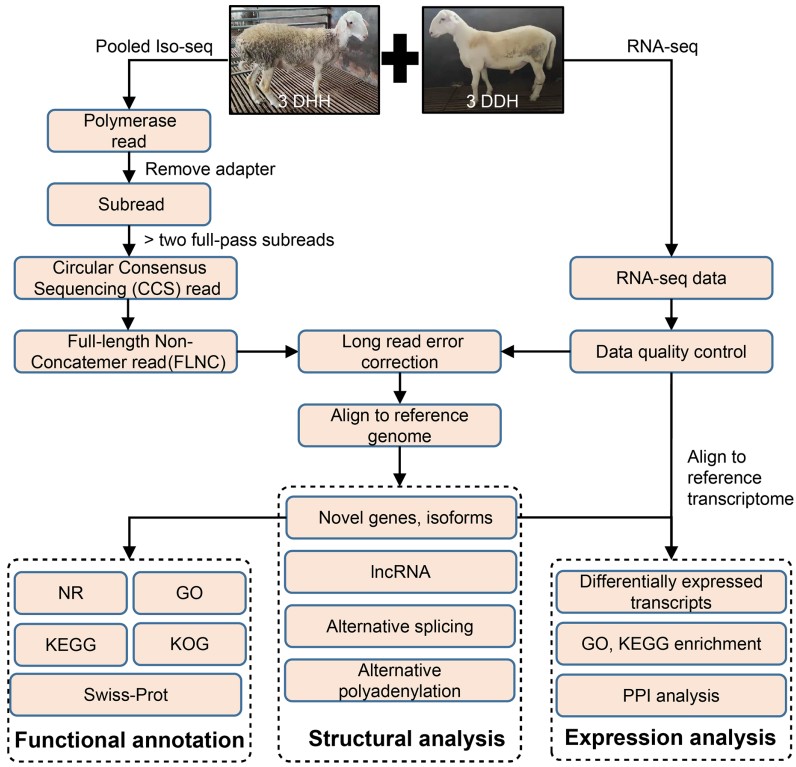

**Figure 1  Overview of bioinformatics pipeline.** DDH, Dorper × (Dorper × Hu sheep); DHH, ((Dorper × Hu) × Hu sheep).                               

analysis. Isoforms meet one of the following conditions were kept: (1) isoform was supported by at least two FLNCs; (2) isoform was supported by one FLNC whose Percentage Identity (PID) was greater than 99%; (3) all splicing sites in an isoform were fully supported by short reads; (4) all splicing junctions in an isoform were annotated by the reference genome. Isoforms overlapped over 20% of their length on the same strand were regarded as transcribing from the same gene locus (*Wang et al., 2021*). A gene locus was defined as a novel gene if it overlaps less than 20% of length with known genes. A novel isoform was defined as an isoform with a final splice site of 3′ ends changed or a new intron (exon) emerged.

## Functional annotation of isoforms

To better understand the potential function of isoforms, they were aligned to Gene Ontology (GO) (*Ashburner et al., 2000*), Swiss-Prot (*Gasteiger, Jung & Bairoch, 2001*), NCBI non-redundant proteins (NR) and Cluster of Orthologous Groups of proteins (COG/KOG) (*Tatusov et al., 2003*) database using Diamond (*Buchfink, Xie & Huson, 2015*). Isoforms were also aligned to KEGG (https://www.genome.jp/kegg/) database using KOBAS (*Xie et al., 2011*).

## lncRNA prediction

Novel isoforms with lengths greater than 200 nt were blast against the NR, KOG/KO and Swiss-Prot to remove transcripts with coding potential. A transcript with above annotated

information was removed for further lncRNA prediction. Multiple software, including CPC 2.0 beta (default parameter) (*Kang et al., 2017b*), CNCI v2 (−m ve) (*Sun et al., 2013*), PLEK v1.2 (−minlength 200) (*Li, Zhang & Zhou, 2014*) and CPAT v1.2.4 (default parameter) (*Wang et al., 2013*), were used to predict lncRNAs from novel isoforms.

### APA and AS analysis

The APA sites for each gene loci were detected using the TAPIS pipeline (*Abdel-Ghany et al., 2016*). AStalavista v3.2 software was used to detect AS events with default parameters (*Foissac & Sammeth, 2007*).

### DETs detection

Reference genome annotation file and PacBio novel isoforms were merged to make a novel annotation file. The clean reads of each RNA-Seq library were aligned to the novel annotation file using the Bowtie2 software (-q −sensitive −dpad 0 −gbar 99999999 −mp 1,1 −np 1 −score-min L,0,-0.1 -I 1 -X 1000 −no-mixed −no-discordant -p 6) (*Langmead & Salzberg, 2012*). The RSEM software was used to quantify transcripts (*Li & Dewey, 2011*). DESeq2 R package (*Love, Huber & Anders, 2014*) was used to identify DETs between thin- and fat-tailed sheep samples. Transcripts with the false discovery rate (FDR) smaller than 0.05 and fold change greater than two were defined as significant DETs. The correlation between differentially expressed lncRNAs and DETs which were transcribed from tail fat related candidate genes was implemented using the ggcor R package (*Huang et al., 2020*).

### Functional annotation of DETs

To extract the potential function of DETs, DETs were blast to GO database using Diamond (*Buchfink, Xie & Huson, 2015*) and KEGG database using KOBAS (*Xie et al., 2011*). FDR < 0.05 was regarded as the significant threshold. The GO enrichment results were visualized by the GOplot 1.0.2 R package (*Walter, Sanchez-Cabo & Ricote, 2015*).

To explore the interaction between DETs and transcripts which were transcribed from the candidate genes linked with tail fat deposition, protein-protein interaction (PPI) networking analysis was implemented using the STRING database (*Szklarczyk et al., 2017*). The result of PPI was visualized by the Cytoscape v3.8.3 software (*Shannon et al., 2003*).

## RESULTS

### PacBio Iso-Seq and bioinformatics analysis

In total, 487,822 polymerase reads were generated by Iso-Seq (Table 2). After quality control, 30,305,694 filtered subreads with a mean length of 1,474 bp and 360,901 CCS reads with an average depth of 71 passes were produced (Table 2). Finally, 271,718 FLNCs with an average length of 1,754 bp were used for further analysis (Table 2).

### Gene loci and isoforms detection

All FLNCs were corrected for sequencing errors and alignment position. According to the alignment position of each FLNC, gene loci and isoforms were identified. In total, 20,041 gene loci (11,040 known gene loci and 9,001 potential novel gene loci) were

**Table 2 Summarized information of Iso sequencing (Iso-Seq) data.**

| Data type | Total number | Min length | Max length | Average length |
|---|---|---|---|---|
| Polymerase reads | 487,822 | 51 | 335,695 | 96,474 |
| Subreads | 30,305,694 | 51 | 250,276 | 1,474 |
| Circular consensus sequencing (CCS) reads | 360,901 | 62 | 15,123 | 2,107 |
| Full-length non-chimeric reads (FLNCs) | 271,718 | 50 | 8,211 | 1,754 |

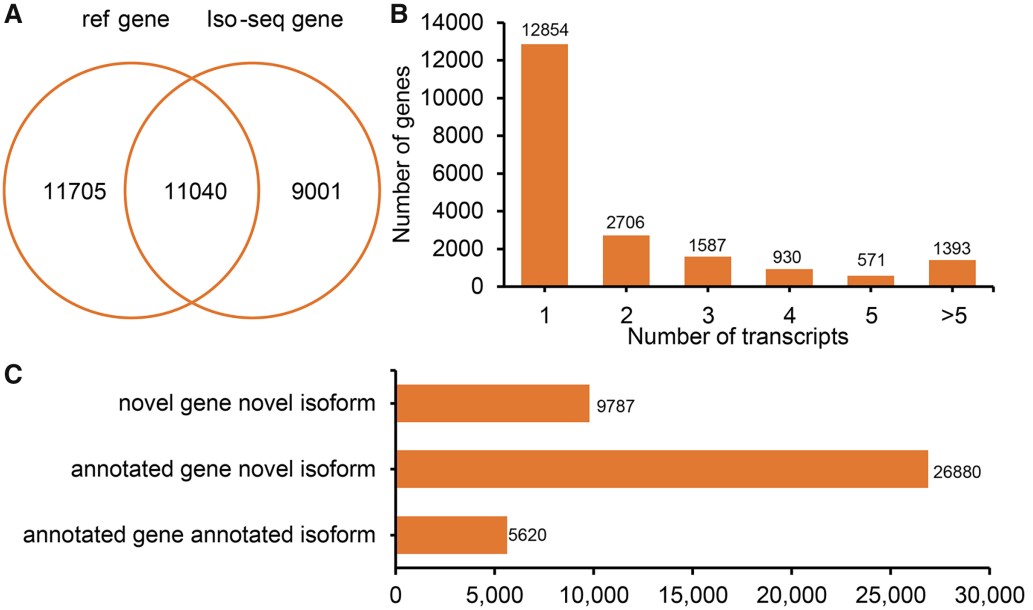

**Figure 2 Genes and isoforms identified by Iso-seqencing (Iso-Seq) data.** (A) Venn plot of annotated genes and identified genes by Iso-Seq data. (B) Distribution of transcribed transcripts. (C) Hist plot of three type of transcripts.

identified by Iso-Seq (Fig. 2A). A total of 7,187 (35.86%) detected genes generate two or more isoforms (Fig. 2B). Interestingly, the *LIPE* (lipase E, hormone-sensitive type) transcribed the greatest number of transcripts (113 novel transcripts and three known transcripts). In sheep tail fat, the majority of novel isoforms are transcribed from annotated genes (Fig. 2C). All novel gene loci and novel isoforms were added to the reference annotation file (Generic Feature Format, GFF, Table S1).

To elucidate the potential function of novel isoforms transcribed from novel genes, isoforms were annotated to public databases, including Gene Ontology (GO) (*Ashburner et al., 2000*), Swiss-Prot (*Gasteiger, Jung & Bairoch, 2001*), NCBI non-redundant proteins (NR), Cluster of Orthologous Groups of proteins (COG/KOG) (*Tatusov et al., 2003*) and Kyoto Encyclopedia of Genes and Genomes (KEGG, https://www.genome.jp/kegg/) database (Table S2). In total, above half novel transcripts (5,762, 58.87%) were annotated at least one public database, including 5,697 (58.21%) in NR, 2,390 (24.42%) in GO, 3,944 (40.30%) in KEGG, 351 (3.59%) in KOG and 1,955 (19.98%) in Swiss-Prot (Fig. 3).

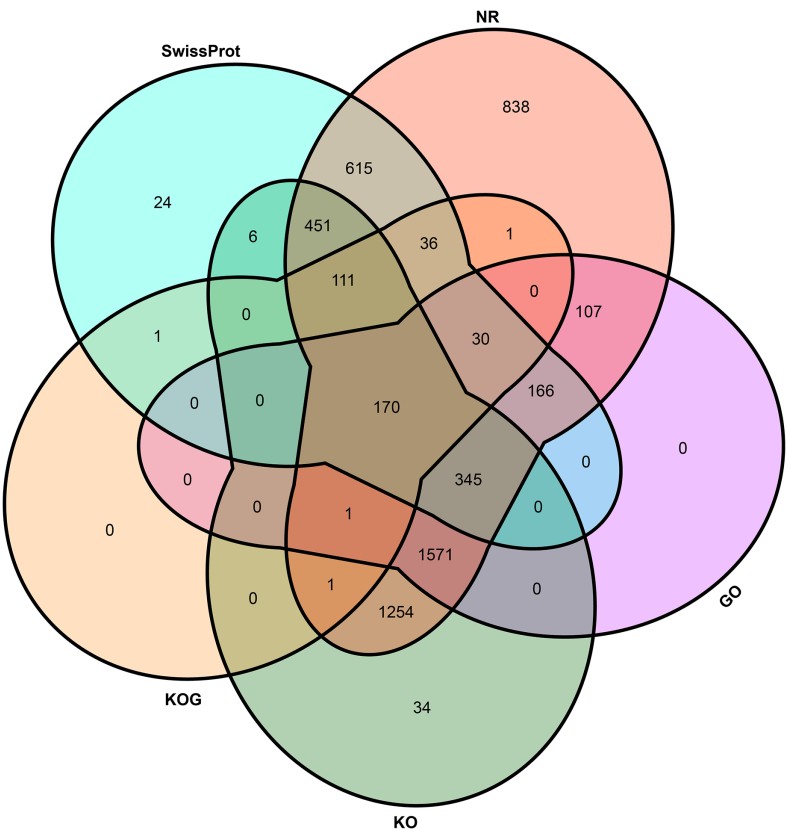

**Figure 3 Venn plot of annotated transcripts.** NR, NCBI non-redundant proteins; GO, gene ontology; KO, KEGG orthology; KOG, cluster of orthologous groups of proteins.

## lncRNA prediction

In the current study, four software were used to predict lncRNAs. Finally, 3,764 lncRNAs were predicted by all software (Fig. 4A). The sequence of predicted lncRNAs were documented in Table S3. The length of predicted lncRNAs ranged from 205 to 5,698 bp (Fig. 4B). The mean length of lncRNAs was 1,198 bp.

## Alternative polyadenylation events detection

TAPIS pipeline was used to detect APA events in tail fat. In total, 5,791 detected genes had at least one poly(A) site (Fig. 5A, Table S4). Of them, 4,129 genes contained a single poly(A) site (Fig. 5A). The remaining 1,662 (28.70%) genes contained two or more poly(A) sites (Fig. 5A). The ENSOARG00020025751 (*ASPH*) and ENSOARG00020017413 (unannotated) contained the greatest number of distinct poly(A) sites, which were illustrated in Figs. 5B and 5C.

## Alternative splicing events detection

In total, 17,834 AS events were detected by Iso-Seq data. AS events were classified into four basic types and others (Fig. 6). Among four basic types, exon skipping (4,679, 26.24%) is

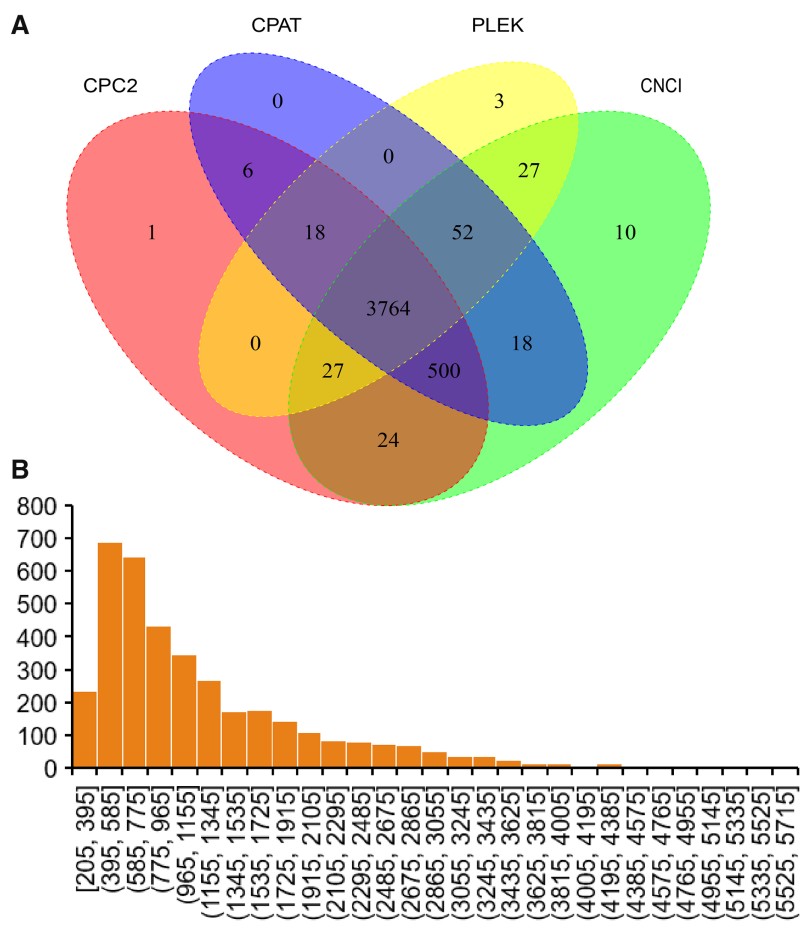

**Figure 4 Identified lncRNAs.** (A) Veen plot of predicted lncRNAs by four software. (B) Length distribution of predicted lncRNAs.

the most prevalent AS event, followed by alternate acceptor site (2,728, 15.30%), intron retaining (2,624, 14.71%) and alternate donor site (1,409, 7.90%).

## Differentially expressed transcripts

In our study, a total of 464 DETs between thin- and fat-tailed samples were identified (Fig. 7A, Table S5). The most significant differential transcript was ENSOART00020021670 (elongation factor 1-beta, FDR = 5.79e-19). Among these DETs, 21 DETs are differentially expressed lncRNAs, including 14 known lncRNAs and seven novel lncRNAs. The correlation between differentially expressed lncRNAs and DETs which transcribed from tail fat related candidate genes, including diacylglycerol O-acyltransferase 2 (*DGAT2*) (*Bakhtiarizadeh & Alamouti, 2020*), acetyl-CoA carboxylase alpha (*ACACA*) (*Bakhtiarizadeh & Alamouti, 2020*; *Bakhtiarizadeh & Salami, 2019*), ATP citrate lyase (*ACLY*) (*Bakhtiarizadeh & Alamouti, 2020*), fatty acid synthase (*FASN*) (*Bakhtiarizadeh & Alamouti, 2020*), stearoyl-CoA desaturase (*SCD*) (*Kang et al., 2017a*), and acyl-CoA synthetase short chain family member 2 (*ACSS2*) (*Guangli et al., 2020*), was investigated. The result suggests that 20 out of 21 lncRNAs (except lncRNA

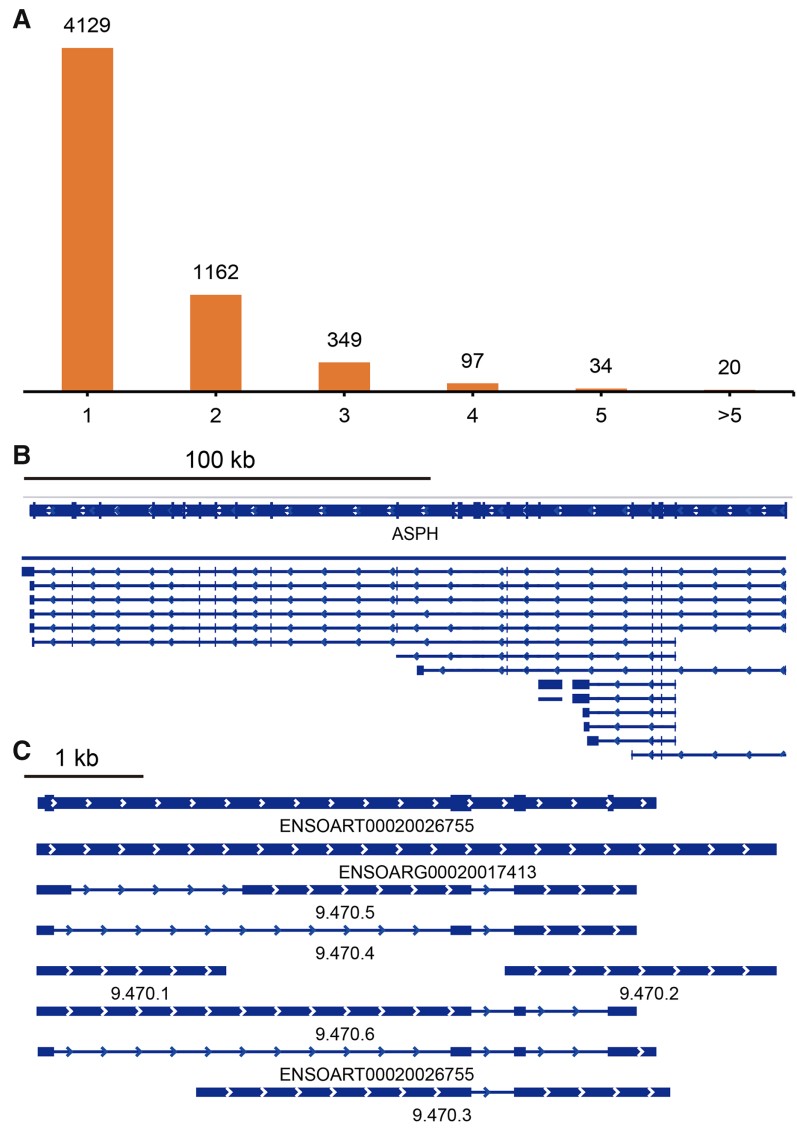

**Figure 5 Identified alternative polyadenylation (APA) events.** (A) Distribution of the number of APA events. (B) APA events in ENSOARG00020025751 (ASPH). (C) APA events in ENSOARG00020017413 (unannotated).

ENSOART00020017088) were significantly correlated with at least one transcripts (Fig. 7B).

## Functional annotation of DETs

PPI analysis was implemented to identify functional similarities between the DETs. All of the networks were documented in Table S6. Here, we highlight an example. *ACACA* is a candidate gene related to tail fat deposition (*Bakhtiarizadeh & Alamouti, 2020*; *Bakhtiarizadeh & Salami, 2019*). In the current study, we found ten DETs (11.99.3, 11.689.28, 1.689.18, 11.265.3, ENSOART00020040904, 11.99.15, 11.689.14, 13.624.23, ENSOART00020006006, and 15.327.19) directly interactives with ENSOART00020037575 which was transcribed from *ACACA* (Fig. 8A). To further

| | | Number of alternative events | Ratio |
|---|---|---|---|
| Exon skipping (ES) | | 4679 | 26.24% |
| Alternate acceptor site (AA) | | 2728 | 15.30% |
| Alternate donor site (AD) | | 1409 | 7.90% |
| Intron retained (IR) | | 2624 | 14.71% |
| Other | | 6394 | 35.85% |

**Figure 6 Classification of alternative splicing events.**

understand the function of DETs, GO and KEGG enrichment analyses were implemented. FDR < 0.05 was regarded as the significant threshold. All DETs were significantly enriched in 38 GO terms (Fig. 8B, Table S7). The most significant enriched GO term is extracellular matrix structural constituent (GO:0005201, FDR = 1.87e-4, Fig. 8B), including three annotated transcripts (ENSOART00020027615, ENSOART00020019556, and ENSOART00020033210) and nine novel transcripts (11.236.9, 10.185.4, 4.380.12, 4.380.22, 10.185.6, 24.240.21, 3.1134.2, 2.407.17, and 10.185.7). Detected DETs were significantly enriched in six KEGG pathways (Fig. 8C, Table S8). The most significant enriched pathway is Extracellular Matrix-receptor (ko04512), including three known transcripts (ENSOART00020039107, ENSOART00020027615, and ENSOART00020006511) and 11 novel transcripts (4.380.22, 11.236.16, 3.143.5, 10.185.4, 10.185.6, 10.185.7, 4.380.12, 12.528.5, 1.2008.6, 1.2007.14, and 11.236.9). The most significant GO term and KEGG pathway are extracellular matrix related indicating extracellular matrix play an important role in tail fat deposition.

## DISCUSSION

In the current study, PacBio Iso-Seq was applied to uncovering the complexity of the transcriptome profile of the tail fat. Using Iso-Seq, 9,001 potential novel genes and 36,777 novel transcripts were detected (Fig. 2). The most notable gene was *LIPE* (lipase E, hormone-sensitive type) with 116 transcripts (113 novel transcripts and three known transcripts). The main function of hormone-sensitive lipase in adipose is to break triacylglycerol into fatty acids (*Recazens, Mouisel & Langin, 2020*). In humans, four exons (called exon B, A, T2 and T1) are alternatively used to produce different *LIPE* transcripts (*Recazens, Mouisel & Langin, 2020*). Similar AS patterns have also been found in rats, ewe, chickens, and fish (grass carp) (*Bonnet et al., 1998*; *Cecilia et al., 1988*; *Sun*

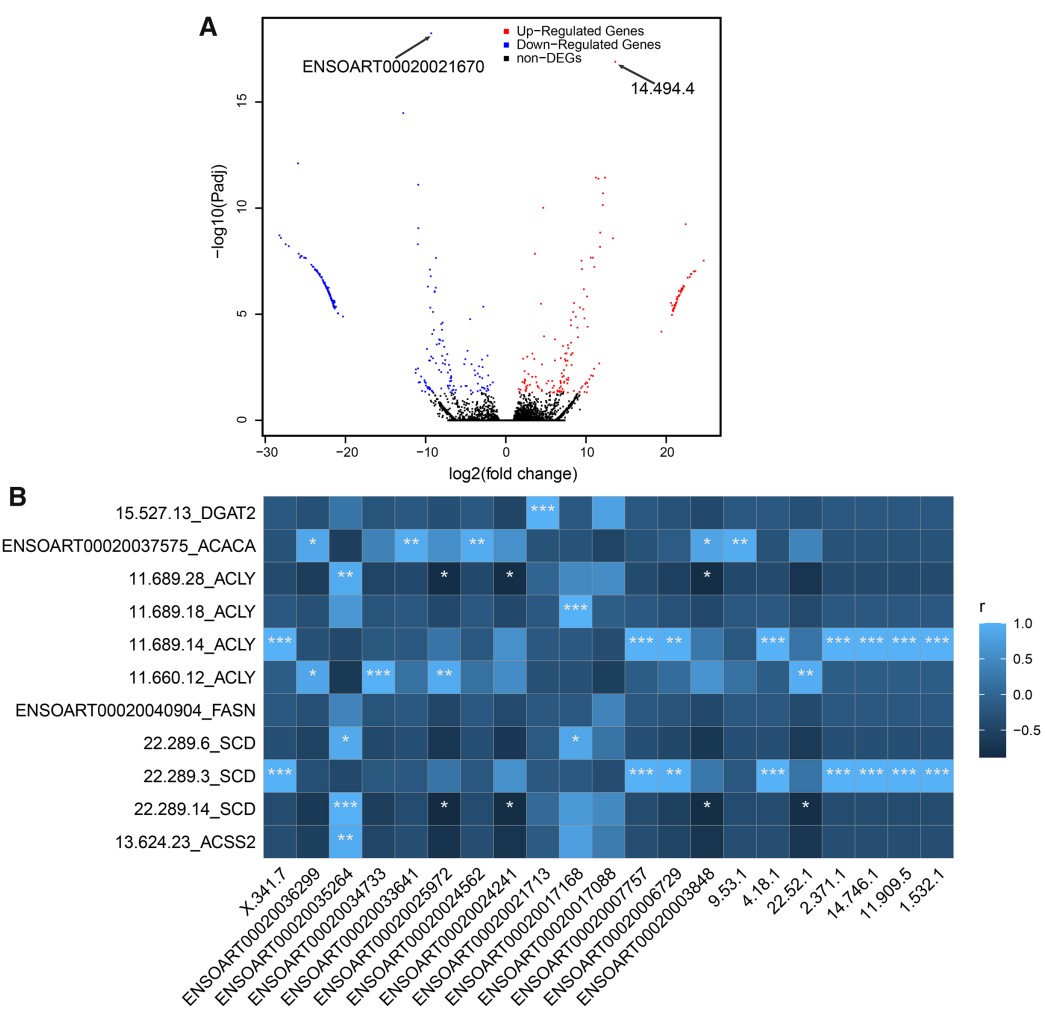

**Figure 7 Identified differentially expressed transcripts (DETs).** (A) Volcano plot of DETs. (B) Correlation of differentially expressed lncRNAs (xlab) and transcripts transcribed from tail fat related candidate genes (ylab). "*" denotes $P < 0.05$. "**" denotes $P < 0.01$. "***" denotes $P < 0.001$.

*et al., 2017*). This suggests that Iso-Seq is promising in identifying novel gene loci and novel isoforms.

LncRNAs were predicted using Iso-Seq data to understand the transcriptome complexity of sheep tail fat. There are previous reports regarding lncRNA regulating tail fat deposition (*Bakhtiarizadeh & Salami, 2019*; *Ma et al., 2018*). These previous studies suggest that only a small portion of novel lncRNA identified in sheep tail fat were conserved (*Bakhtiarizadeh & Salami, 2019*). In the current study, we identified 3,764 lncRNA (Fig. 4, Table S3) and they could be useful for further investigating how lncRNA regulates tail fat deposition. Among them, 21 differentially lncRNAs were identified (Fig. 7B). It has been reported that lincRNA.3,473 interactive with *ACACA* may regulate fat deposition (*Bakhtiarizadeh & Salami, 2019*). In the current study, the positive significant correlation between several lncRNAs (ENSOART00020036299, ENSOART00020033641, ENSOART00020024562, ENSOART00020003848, and 9.53.1)

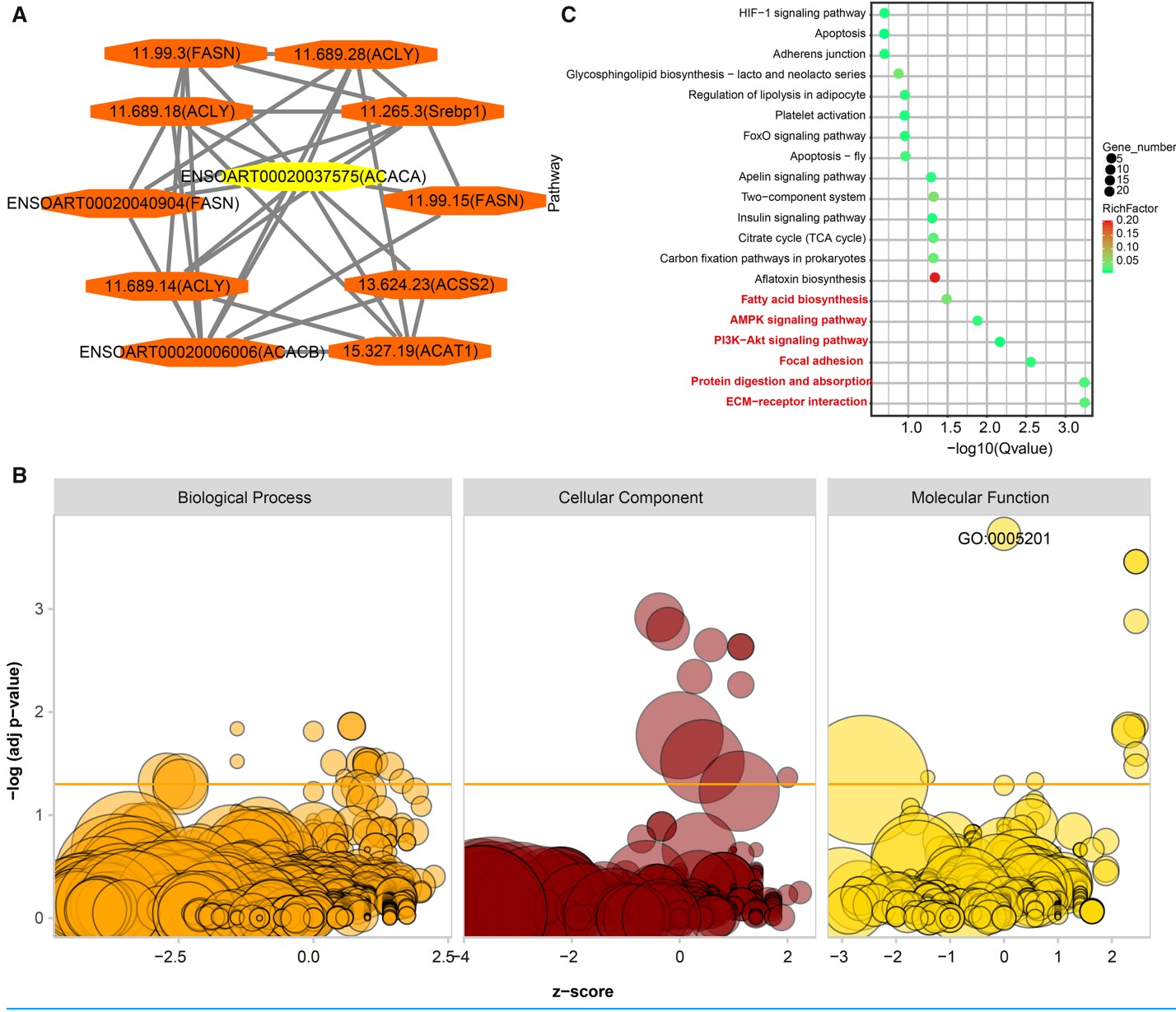

**Figure 8 Functional annotation of differentially expressed transcripts (DETs).** (A) DETs interative with ENSOART00020037575 (transcribed from ACACA). (B) Gene Ontology (GO) enrichment analysis of DETs. (C) Kyoto Encyclopedia of Genes and Genomes (KEGG) enrichment analysis of DETs.

and *ACACA* indicates that these lncRNAs may play a key role in tail fat deposition. The result that the majority of differential expressed lncRNAs (except lncRNA ENSOART00020017088) were significantly correlated with one or more transcripts transcribed from tail fat linked candidate genes could further validate their potential function in tail fat deposition. However, the genetic mechanism underlying lncRNAs regulate tail fat deposition still needs to be further investigated. In addition, APA was also investigated. In total, 1,662 genes contained at least two poly(A) sites were detected in sheep tail fat (Fig. 5). APA-mediated gene regulatory functions are tissue-specific

(*Di Giammartino, Nishida & Manley, 2011*; *Lianoglou et al., 2013*; *MacDonald, 2019*; *Tian & Manley, 2017*). Generally, APA could affect mRNA stability, splicing, localization and translation by alter the sequence of regulatory elements, *e.g.*, miRNA binding sites (*Elkon, Ugalde & Agami, 2013*; *Tian & Manley, 2017*). Although how the APA regulates tail fat deposition still needs further study, our results would be a useful resource for further analysis the function of APA in sheep tail fat.

The main function of AS is to increase the diversity of transcripts (*Keren, Lev-Maor & Ast, 2010*). As a result, transcripts expression abundance may change with different phenotypes. Using integrative analysis of Iso-Seq and RNA-seq data, hundreds of DETs were identified between thin- and fat-tailed sheep (Fig. 7A). One of DETs 15.527.13 was transcribed from *DGAT2*. *DGAT2* has been reported to affect tail fat deposition in sheep (*Bakhtiarizadeh & Alamouti, 2020*) and back fat deposition in pig (*Yin et al., 2012*). A novel transcript 13.624.23, also a DET, was transcribed from *ACSS2*. Four DETs, 11.689.28, 11.689.18, 11.689.14, and 11.660.12, were transcribed from *ACLY*. *ACSS2* and *ACLY* have been identified as candidate genes related to tail fat deposition in sheep (*Bakhtiarizadeh & Alamouti, 2020*; *Guangli et al., 2020*). Cytosolic acetyl CoA (Ac-CoA), the central precursor for lipid biosynthesis in mammals, is produced by *ACLY* from mitochondria-derived citrate or by *ACSS2* from acetate (*Vysochan et al., 2017*). ENSOART00020040904 was transcribed from *FASN*. Three DETs (22.289.6, 22.289.3 and 22.289.14) were transcribed from *SCD*. ENSOART00020037575 was transcribed from *ACACA*. *FASN*, *ACACA* (*Bakhtiarizadeh & Alamouti, 2020*) and *SCD* (*Kang et al., 2017a*) are candidate genes related to tail fat deposition. These three gene, *FASN* (*Leonard et al., 2004*), *SCD* (*Garcia-Fernandez et al., 2010*; *Kang et al., 2017a*) and *ACACA* affect fatty acid composition (*Cronan & Waldrop, 2002*). These suggest that many novel transcripts were transcribed from well-studied candidate genes linked to lipid synthesis and may regulate tail fat deposition, highlighting alternative splicing is a complicated process in regulating tailed fat deposition.

PPI analysis is an effective way to search functional similarities between the DETs. For example, ENSOART00020037575 (transcribed from *ACACA*) directly interactives with 11.265.3 (transcribed from sterol regulatory element binding transcription factor 1, S*rebp1*) implying that *Srebp1* may be linked with tailed fat deposition (Fig. 8A). *Srebp1* has been reported to affect tail traits and lipid metabolism in sheep (*Liang et al., 2020*), which could validate our predicted result. In addition to PPI analysis, the functionality of DETs were also been investigated by GO and KEGG enrichment analyses. Interestingly, the most significant GO term (GO:0005201, extracellular matrix structural constituent, Fig. 8B, Table S7) and KEGG pathway (ko04512, ECM-receptor interaction, Fig. 8C, Table S8) is extracellular matrix related. Previous studies suggest that differentially expressed genes (DEGs) between fat-tail tissue of fat-tailed sheep and thin-tailed sheep were significantly enriched in the ECM-receptor interaction pathway (*Bakhtiarizadeh et al., 2019*; *Li et al., 2018a*). The studies in cattle and human adipose tissue also suggest that ECM plays an important role in adipogenesis (*Casado-Diaz et al., 2017*; *Lee et al., 2013*). Consistent with these results, in the present study, many novel DETs were

significantly enriched in ECM-related GO terms and pathways, which suggests that Iso-Seq combined with the RNA-seq is a useful way to identification DETs.

## CONCLUSIONS

In the present study, PacBio Iso-Seq was used for producing comprehensive transcriptomic data of tail fat of sheep, result in 9,001 potential novel gene loci, 17,834 AS events, 5,791 APA events and 3,764 lncRNAs. Combined with Iso-Seq and RNA-Seq data, hundreds of DETs between thin- and fat-tailed sheep were identified. Among them, 21 differentially expressed lncRNAs, such as ENSOART00020036299, ENSOART00020033641, ENSOART00020024562, ENSOART00020003848 and 9.53.1 may regulate tail fat deposition. Many novel transcripts were identified as DETs, including 15.527.13 (*DGAT2*), 13.624.23 (*ACSS2*), 11.689.28 (*ACLY*), 11.689.18 (*ACLY*), 11.689.14 (*ACLY*), 11.660.12 (*ACLY*), 22.289.6 (*SCD*), 22.289.3(*SCD*) and 22.289.14 (*SCD*), highlighting AS is a complicated process in regulating tailed fat deposition. Identified DETs were most significantly enriched in ECM-related GO terms and KEGG pathways suggesting their important roles in lipid metabolism. Our result revealed the transcriptome complexity and identified many candidate transcripts in tail fat, which could enhance the understanding of the mechanisms of tail fat deposition.

### Funding

This research was funded by the National Natural Science Foundation of China (31872333, 32172689), National Natural Science Foundation of China-CGIAR (32061143036), Major New Varieties of Agricultural Projects in Jiangsu Province (PZCZ201739), The Projects of Domesticated Animals Platform of the Ministry of Science, Key Research and Development Plan (modern agriculture) in Jiangsu Province (BE2018354), Jiangsu Agricultural Science and Technology Innovation Fund (CX(18) 2003), and Natural Science Foundation of Jiangsu Province (BK20210811). The funders had no role in study design, data collection and analysis, decision to publish, or preparation of the manuscript.

### Grant Disclosures

The following grant information was disclosed by the authors:
National Natural Science Foundation of China: 31872333 and 32172689.
National Natural Science Foundation of China-CGIAR: 32061143036.
Major New Varieties of Agricultural Projects in Jiangsu Province: PZCZ201739.
The Projects of Domesticated Animals Platform of the Ministry of Science.
Key Research and Development Plan (modern agriculture) in Jiangsu Province: BE2018354.
Jiangsu Agricultural Science and Technology Innovation Fund: CX(18)2003.
Natural Science Foundation of Jiangsu Province: BK20210811.

## Competing Interests

The authors declare that they have no competing interests.

## Author Contributions

- Zehu Yuan conceived and designed the experiments, performed the experiments, analyzed the data, prepared figures and/or tables, authored or reviewed drafts of the paper, and approved the final draft.
- Ling Ge performed the experiments, analyzed the data, prepared figures and/or tables, and approved the final draft.
- Jingyi Sun performed the experiments, analyzed the data, prepared figures and/or tables, and approved the final draft.
- Weibo Zhang performed the experiments, prepared figures and/or tables, and approved the final draft.
- Shanhe Wang performed the experiments, prepared figures and/or tables, and approved the final draft.
- Xiukai Cao performed the experiments, prepared figures and/or tables, and approved the final draft.
- Wei Sun conceived and designed the experiments, performed the experiments, analyzed the data, prepared figures and/or tables, authored or reviewed drafts of the paper, and approved the final draft.

## Animal Ethics

The following information was supplied relating to ethical approvals (*i.e.*, approving body and any reference numbers):

Animal experiment were approved by the Experimental Animal Ethical Committee of Yangzhou University (NO.202103294).

## Data Availability

The data is available at NCBI SRA: PRJNA745517.

## Supplemental Information

Supplemental information for this article can be found online at http://dx.doi.org/10.7717/peerj.12454#supplemental-information.

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
