# Peer review of "Integrative analysis of Iso-Seq and RNA-seq data reveals transcriptome complexity and differentially expressed transcripts in sheep tail fat"

_PeerJ, doi:10.7717/peerj.12454_

## Round 0.1 · original submission · Major Revisions

The information about controls is lacking. It would have been meaningful if authors would have used pure breds of both HU and dorper breeds as controls for meaningful comparative results.

Background and method section in the abstract is inadequate.

The method section should be clearly written with complete details of methodology followed. For scientific integrity all data should be deposited in NCBI.

Since RNA is unstable, how has the tissue been directed freezed?

Give the details about the weight and feeding regime of animals. I am also interested to know the thickness or weight of fat per unit area in the tail region.

As written by the authors, it is important to look for candidate genes responsible for fat deposition. I wonder, I couldn’t find any experiments related to it. If candidate genes are already identified, I suggest going for protein-protein interaction/ networking of top expressed genes with candidate genes already available.

How many LnCRNA are novel to fat deposition?. A comparative analysis is required.

To predict lncRNA with greater certainty, I recommend to use additional algorithms, like CNCI, PFAM and PLEK,

Correlation analysis between lncRNAs and genes related to fat deposition should also be carried out.

Conclusion is vague, should be refined.

Reviewer 1 ·

Basic reporting

The authors studied the tail fat content trait in cross-breed Hu sheep (DHH as fat-tail and DDH as thin-tail). Transcriptome analysis was performed using long reads (generated by PacBip platform) and short reads and authors tried to identify novel splicing variants, lncRNAs, and differentially expressed genes/transcripts. They also investigate alternative polyadenylation events.

Experimental design

All of my comments could be found in the attached pdf file. My edits are also highlighted in track changes.

Validity of the findings

There are a few issues regarding validity:
1- First, you said you identified 9001 novel gene loci (Line 188) but you didn't give any proof that they are really novel genes. You must either provide evidence for this claim or rewrite the sentence.
2- The authors performed differential expression analysis and reported DEGs/DETs but didn’t perform qRT-PCR for validation. Transcriptome analysis definitely should be accompanied by a validation qRT-PCR. Please consider it.

Additional comments

1- Methods are not well-detailed. More details should be added.
2- Please check all abbreviations and give explanations in the first appearance. Also, give explanations for all abbreviations in the tables and figures. tables should be self-informative.

Annotated reviews are not available for download in order to protect the identity of reviewers who chose to remain anonymous.

·

Basic reporting

The authors used RNA-Seq to generate a transcriptomic data of tail fat tissue of sheep.
Novel transcripts in sheep tailed fat were identified. Differential expressed transcripts (DETs) between thin- and fat-tailed sheep were identified. GO and KEGG enrichment analysis were implemented to investigate the potential function of DETs. The big concern is about the used methods for aligning the reads against genome as well as lncRNA prediction method, which is not robust enough. Also, discussion is not appropriate and have to be enriched.

Experimental design

The manuscript needs to be revised by considering the following recommendations.

1- Line 35: Authors claimed that for the first time they revealed the transcriptomic complexity of the tail fat of sheep!!! There are several studies in this filed with the same subject.

2- Why only one sample was sequenced by Iso-Seq? As it is obvious isoforms and transcripts are different between thin and fat-tailed sheep. When all samples are pooled, how can you identify specific isoforms. In other words, how is it possible to differentiate isoforms/transcripts of each treat? Although Iso-Seq can sequence long transcripts, but in this study, authors have to sequence at least one sample per treat to be sure about the results.

3- The filtering pipeline for lncRNA perdition is a little bit easy going and only CAPT software is used. For example, always several coding potential prediction tools have to be used and the results of all tools consider. Also, open reading frame prediction is the other step to remove some transcripts with coding potential. Why did not use these steps?

4- Bowtie2 is used to align the reads against the genome? Why? As the data is RNA-Seq, a splice aware aligner, such as HISAT, have to be used.

5- Differentially expressed mRNAs were not discussed and the results of their enrichment analysis are not provided. In discussion different aspects of the results have to be interpreted. I think that discussion need to be enriched.

6- RNA-Seq samples have to be deposited in available databases like NCBI and the accession number of the data be mentioned in the manuscript.

7- There are assorted typos and grammatical errors that need to be corrected. Totally, the manuscript has to be revised by a native speaker. For example:
a. Line 117: “(Sage Science, MA, USA) was use …” have to be “(Sage Science, MA, USA) was used”
b. Line 120: “(Life Technologies, CA, USA) was used” have to be “(Life Technologies, CA, USA) were used”

Validity of the findings

The results of RNA-Seq did not validate by Real-Time PCR, as it is needed.

---

## Round 0.2 · accepted · Accept

I recommend the manuscript for publication in PeerJ.

Reviewer 1 ·

Basic reporting

Dear Authors,

Thanks for revising the manuscript according to my comments. The majority of the raised points are addressed properly. I am still not totally convinced about the number of newly identified genes/transcripts. However, in the revised version, the limitations of the study were acknowledged, and the extra information provided in the tables and the supplementary file could also help the readers to decide about the soundness of the results.
The manuscript reads better and the English language is acceptable but it is not perfect. Given the limited number of published papers in sheep transcriptomics, I am happy to endorse the manuscript for publication in PeerJ.

Experimental design

No comment.

Validity of the findings

No comment.

·

Basic reporting

After checking the original text and its response carefully, I think that the current quality of the paper is suitable for publication in peer J. This manuscript is written in a standard manner, scientific issues are explained clearly.

Experimental design

There are indeed some problems with the experimental design. The fat tail trait is a complex quantitative trait. It mainly includes two main traits: fat deposition trait and long and short tail trait. Hu sheep is a short fat-tailed sheep breed, while Dorper sheep is a long thin-tailed sheep breed. Therefore, I think it would be more meaningful if choosing a short thin-tailed sheep breed such as Tibetan sheep as a control in this study.It is more scientific for studying fat trait.

Validity of the findings

The results of the research data are relatively reliable, and the analytical methods are appropriate, so the conclusions obtained are relatively reliable.

Additional comments

N.A.